# Evaluating Community Capability to Prevent and Control COVID-19 Pandemic in Shenyang, China: An Empirical Study Based on a Modified Framework of Community Readiness Model

**DOI:** 10.3390/ijerph20053996

**Published:** 2023-02-23

**Authors:** Xiaojie Zhang, Xiaoyu Liu, Lili Wang

**Affiliations:** 1Department of Public Administration, School of Humanities & Law, Northeastern University, Shenyang 110169, China; 2Party School of Weihai Municipal Committee of Communist Party of China, Weihai 264213, China

**Keywords:** community capability, COVID-19, community readiness model, community attachment, China

## Abstract

Community plays a crucial role in the successful prevention and control of the COVID-19 pandemic in China. However, evaluation of community capability to fight against COVID-19 is rarely reported. The present study provides a first attempt to assess community capability to combat COVID-19 in Shenyang, the capital city of Liaoning province in Northeast China, based on a modified framework of a community readiness model. We conducted semi-structured interviews with ninety key informants from fifteen randomly selected urban communities to collect the data. The empirical results indicate that the overall level of community capability for epidemic prevention and control in Shenyang was at the stage of preparation. The specific levels of the fifteen communities ranged from the stages of preplanning to preparation to initiation. Concerning the level of each dimension, community knowledge about the issue, leadership, and community attachment exhibited significant disparities between communities, while there were slight differences among communities on community efforts, community knowledge of efforts, and community resources. In addition, leadership demonstrated the highest overall level among all the six dimensions, followed by community attachment and community knowledge of efforts. Community resources displayed the lowest level, followed by community efforts. This study not only extends the application of the modified community readiness model to evaluate community capability of epidemic prevention in the Chinese community context, but also offers practical implications for enhancing Chinese communities’ capabilities to deal with various future public health emergencies.

## 1. Introduction

The coronavirus disease 2019 (COVID-19) pandemic, which first broke out at the end of 2019 in China [1], was called a “public health emergency of international concern” by the World Health Organization because of its high infection and mortality rate [2]. It has caused global social disruption and economic recession and continues to pose a major threat to public health. As of 2 December 2022, over 639 million people have been infected; of these, 6.6 million deaths, an unprecedented rate of spread around the world [3]. While the losses of the pandemic are numerous, many countries are still struggling to prevent and control the COVID-19 disease. China was the first country to identify and report a confirmed case, and it has effectively controlled the wide spreading of the pandemic through strict and effective measures since the first outbreak of COVID-19 in Wuhan of China up until the Chinese government deregulated the epidemic control in December 2022. The number of deaths due to the infection of COVID-19 is fewer than 32,000 in China since the first outbreak of the pandemic [4].

During the process of the COVID-19 prevention and control in China, communities played a crucial role in implementing government policy, mobilizing public participation, and providing public service. For example, communities set up checkpoints at the entrances of the blockaded communities and provided the necessities of life for residents [5,6], helped organize and conducted large-scale community-based biotechnology testing of COVID-19, offered medical assistance, provided isolation rooms for suspected cases or those who came from the epidemic areas, disinfected public spaces regularly, recruited and trained volunteers, disseminated anti-epidemic knowledge, and provided psychological counseling [7]. A few studies have shown that these measures taken by communities effectively reduced the risk of the COVID-19 outbreaks [8,9,10,11]. However, the performance of communities in the pandemic prevention and control varies widely in China. Some communities can provide efficient and effective preventive services while others cannot. Some communities have offered sufficient and satisfactory residents’ demand-oriented services while other counterparts have not. The variation of the performance is mainly due to the different levels of communities’ preventive and controlling capabilities [12]. As a result, the main questions that this research addresses are (i) How can community capability be evaluated in China? and (ii) What are the levels of community capability for epidemic prevention and control in China?

Community capability, also called community capacity [13], refers to the residents’ ability to collectively affect community opportunities [14]. According to George et al., community capacity refers to the combined influence of a community’s social systems and collective resources, which is generally applied to address community problems and broaden community opportunities [15]. Since the community is not a passive recipient of outside influences but an active initiator in the effort to achieve specific goals [16], Chavis emphasized more the initiative in the defining process. Chavis identified community capacity as ‘‘the ability to develop, mobilize, and use resources to manage change’’ [17]. With the emphasis on community capacity increasing, assessing community capability has emerged rapidly from the fields of community participation and development in recent years. Brock et al. maintained that the development of capability assessment allows the community to understand its strengths and room for further improvement [18].

Previous research has addressed a few assessment frameworks that are used directly or adapted slightly in evaluating community capability. For example, the NSW Health Capacity Building Framework is frequently used to evaluate community capacity in public health management [19,20]. The framework provides a guide for enhancing the capability of the community system to improve health [21]. Goodman et al. constructed a ten-dimension evaluative framework of the community capacity, and subsequent studies further verified the validity of some dimensions and also highlighted the importance of others, such as cultivation of leadership [22,23]. Liu constructed an evaluation index based on Chinese communities’ characteristics, which involves six dimensions, i.e., community participation, community consciousness, horizontal and vertical interaction, leadership, problem assessment ability, and resource mobilization ability [24]. Lee also developed a six-dimension scale for assessing community capacity. It is comprised of leadership and organization, administrative management, resource mobilization, residents’ participation, collaborative work and network, and public relations and initiatives [25].

It is clear that there are some common indices within those evaluative frameworks. They are community leaders who can initiate mobilization, partnerships of community connectedness and concern, the availability of and access to internal and external resources, and linkages and networks that can facilitate collective action [26]. Most of these common indices are incorporated in the community readiness model (hereafter CRM), which is based on structured interview guidelines and scoring systems. The model is widely used to evaluate communities’ preventive ability in drug and alcohol, intimate partner violence, childhood obesity, cardiovascular disease, HIV/AIDS, and cancer prevention [27,28,29,30], demonstrating high validity and wide applicability. Furthermore, the CRM can help theorists and practitioners have a deep understanding of community capability from the perspective of group dynamics [31]. The application of the key informant method inherent in the model contributes to better knowledge of the practical progress of the targeted intervention programs in the communities. In addition, the model provides an easily operationalized measurement tool for scholars and accurate intervention strategies for policymakers to push the community to change.

With regard to studies assessing community readiness for COVID-19 prevention, Adane et al. evaluated community readiness level for COVID-19 pandemic prevention in the Awi Zone of northwest Ethiopia by employing four evaluative criteria including residents’ knowledge, vulnerability perception, attitude, and practice towards prevention measures [32]. Bumyut et al. used the CRM to estimate the community readiness for implementation of the Safety and Health Administration for COVID-19 prevention in the tourism community of Southern Thailand [33].

Previous studies focused mainly on the establishment of evaluation frameworks and the adoption of those frameworks to assess community capability of mobilizing and taking action to prevent chronic disease, serious disease, addictive behavior, domestic violence, and health promotion. Very few studies evaluated community capability to deal with the worldwide COVID-19 epidemic [33]. As far as we know, no studies have been published to evaluate community ability to deal with public health emergencies including COVID-19 in China. Therefore, the main purpose of this study is to estimate community capability to prevent and control the COVID-19 pandemic in China by employing a modified framework of the widely used CRM.

## 2. A Modified Framework of the CRM

The CRM was originally developed by the Tri-Ethnic Center for Prevention Research at Colorado State University to measure a community’s level of readiness to implement a prevention program [34,35]. The aim of constructing this theoretical model is to identify whether a local prevention program can be effectively and successfully carried out and supported by a community, and to offer strategies to help communities’ mobilization for better changes. The original CRM includes five dimensions of readiness: (a) community efforts, (b) community knowledge of the efforts, (c) leadership, (d) community knowledge about the problem, and (e) funding for community efforts [35,36]. In the subsequent development of the theory, the fifth dimension “funding for community efforts” was changed to “resources for community efforts”, in order to incorporate other important resources besides money, including people, time, space, and other factors that also influence community efforts. A sixth dimension called community climate, which refers to characterization of a community, was also added according to the suggestions of community members, who participated in a workshop where the model was presented. These dimensions cover a variety of aspects that can help guide a community in moving their readiness levels forward [29].

The CRM divides community readiness level into nine stages that range from “community tolerance or no awareness of the issue” to “a high level of community ownership”. The nine stages of readiness are (a) community tolerance or no awareness, (b) denial/resistance, (c) vague awareness, (d) preplanning, (e) preparation, (f) initiation, (g) institutionalization or stabilization, (h) confirmation/expansion, and (i) professionalization [27,34]. The specific definition of each stage, which can also be used as an “anchored statement” to evaluate community readiness level, is shown in Table 1. Specifically, stage 1 to stage 9 represents a continuum of low to high level of community readiness to implement a specific program. Stage 1, which corresponds to the anchored scales score of 1, means the lowest level, while stage 9 corresponding to the scales number of 9 means the highest level. Once the stage of community readiness is identified, intervention strategies can be formulated and implemented to raise levels of community readiness [37].

Since the creation of the CRM, it has been extensively used to identify the level of community readiness to develop and implement prevention and treatment programs for addressing a variety of problems ranging from environmental problems such as air and water pollution, litter, and recycling; health and nutritional problems such as obesity, cancer, drug and alcohol abuse, cardiovascular disease, and sexually transmitted diseases; to social problems such as violence, transportation safety, poverty, and homelessness [27,29,30,35,38,39,40,41]. The wide application of this model demonstrates its appropriateness, effectiveness, and high diagnostic power in determining the stages of community readiness and enhancing its level to deal with a wide range of problems. As community readiness is often used interchangeably with community capability [23], and CRM has been shown to be a very effective tool for building community capability [42], we chose to adopt the CRM to evaluate community capability to prevent and control the COVID-19 pandemic in China. We also used community readiness and community capability in the same sense.

With the continuously extended application of the CRM, it has evolved over time and became a flexible organic system that can be adapted to new and different problems [36]. According to Kostadinov et al. [43], in order to better tailor the model to the subject area and particular community, scholars make both minor modifications, including modifying the methodology and interview scripts, and substantial changes, including removing or adding the core questions, changing dimensions, adding new dimensions, and altering existing dimensions. For example, Apriningsih et al. examined the school readiness for the implementation of a school-based Weekly Iron Folic Acid Supplementation Program by integrating the social ecological model into the CRM [44]. York et al. substituted a new political climate dimension for the knowledge of existing efforts dimension [45]. Jason et al. divided the community climate dimension into town climate and police department climate, in order to reflect the differences between two sections of the community [46]. Gansefort et al. combined the community efforts and the community knowledge of the efforts into one single dimension [47]. Bumyut et al. removed the community efforts dimension when evaluating the community readiness for effective implementation of COVID-19 prevention measures [33]. Liu et al. also removed community efforts in assessing community readiness for disseminating evidence-based physical activity programs to older adults [48].

The studies mentioned above indicate that modifications to the CRM are widely accepted and employed to make the model better fit the community, the subject area, and the particular issue. Just as Jumper-Thurman et al. addressed, “the model is a research-based tool” [49]. In this study, we substitute community attachment for the community climate dimension to better tailor the model to the Chinese community and the COVID-19 pandemic issue. The community climate is originally defined as the level of community support for specific programs, such as the opportunities, policies, services or staffs, and so on [50]. In its subsequent application to measure community readiness, it mainly refers to community attitudes [51] and is defined as prevailing attitudes within the community concerning the issue [28]. The measuring questions of community climate focus mainly on community members’ attitudes toward the issue, the primary obstacles to community efforts, community members’ support on efforts to address the issue, and the circumstance in which community members tolerate the issue. Most of the measures are similar to the measures of community knowledge of efforts, resources, and community knowledge, especially in the Chinese context. For example, measurements of community members’ support and the primary obstacles can be covered by the measures of resources, and the measurement of community attitudes can be substituted by the measures of community knowledge. Therefore, we remove the dimension of community climate.

Simultaneously, we add the community attachment dimension. Community attachment refers to the feeling of being part of a group that is a source of security and a kind of emotional connection with the community [52]. Attachment also implies that this sense of belonging is positively evaluated, and that one is happy and proud to belong to the community and will take on more responsibility in the community [53]. Community attachment significantly impacts residents’ participation in community activities of improving its members’ well-being and addressing social needs and other urban issues [54]. A number of prior studies have demonstrated the importance of community attachment in explaining community participation and satisfaction [55,56,57,58,59]. For example, community attachment or the sense of community has been shown to significantly predict residents’ involvement in substance abuse prevention activities [60]. Since community participation is an essential element of community capability, the community attachment is thus added, in order to enhance the applicability, appropriateness, and explanatory power of the CRM to evaluate community capability to prevent and control the COVID-19 pandemic in China.

## 3. Materials and Methods

### 3.1. Design and Sample

In this study, community refers to the jurisdiction of the community neighborhood committees. To assess the level of community capability for COVID-19 prevention and control in China, we selected two communities randomly in each of the nine administrative districts in Shenyang, the capital city of Liaoning Province in Northeast China. The reasons why we chose the city of Shenyang as the targeted research area were as follows.

First of all, Shenyang is the most important central city in Northeast China. It has a population of 9.118 million and is classified as a mega-city region by the Central People’s Government of China. Conducting a case study of Shenyang could provide some community governance experiences and intervention strategies to enhance community capability to cope with various public health emergencies to other similar mega-cities. Secondly, Shenyang has experienced several serious outbreaks of the COVID-19 pandemic since 2020 and the communities in the city have made tremendous efforts in effectively fighting against the epidemic. Therefore, the communities in Shenyang are representative in the implementation of epidemic prevention and control programs. Evaluating the community capability in Shenyang may contribute to a better understanding of a variety of strengths and weaknesses of the communities and their readiness to prevent and control the epidemics.

After choosing the targeted research communities, we adopted the qualitative approach to assess the community capability level of the prevention and control of COVID-19. Key informants’ interview and anchor scoring method were both used to obtain the research data and identify the accurate capability level, the details of which are displayed in Section 3.2 and Section 3.3.

### 3.2. Data Collection

The CRM is used to evaluate a community’s level of readiness through a semi-structured interview of key informants. Key informants are formal and informal community leaders or decisionmakers who can provide comprehensive and informed opinions about various problems in the community. They represent different sectors of communities and have extensive experience of working with their communities. The criteria of key informant selection in this study are as follows: (1) at least one year of work experience in the community neighborhood committees or living in the community for more than three years; (2) deeply involved in the COVID-19 prevention and control; and (3) willing to participate in the study. Six key informants in each community were selected. Finally, ninety key informants from fifteen communities, including Communist Party of China community neighborhood committee secretaries, community workers, community volunteers, and community elites, were interviewed to acquire the qualitative research data. These interviewees had the best understanding of what happened in the communities and what the communities had done to prevent and control the COVID-19 pandemic.

To ensure the quality of the interview, we developed the final interview outline based on the instructions of the Tri-Ethnic Center, and we also conducted some pre-interviews to modify the outline to make it more suitable to the Chinese community context and more comprehensible for the key informants. The final modified interview outline is shown in Table 2.

The semi-structured interviews with the community key informants were conducted from April to June in 2022. Due to the COVID-19 pandemic, all the interviews were conducted by telephone separately. At the beginning of the formal interview, we ensured that all interviewees had a full understanding of the purpose and process of the research, and we also informed all interviewees that the interview was confidential. Moreover, we obtained the electronic signatures of the interview participants on the informed consent form for the use of the research data. Finally, each interview lasted for about thirty to forty-five minutes, and the interviews were fully audio-taped and transcribed verbatim for subsequent analysis.

### 3.3. Data Analysis

Firstly, according to the instructions given by the Tri-Ethnic Center, anchor scoring method was used to evaluate the specific capability level of different communities. Since the statements of the anchor rating scale were rooted in the Western community context, the contextual differences between China and the West may lead to biased evaluation to a certain extent. Thus, we adjusted the anchor sentences according to experts’ advice, the pre-surveys, and Chinese language expressions. For the newly added dimension of community attachment, we developed the anchor sentences based on the studies of Castañeda et al. and Foster-Fishman [61,62], and the instructions of scoring dimensions by the Tri-Ethnic Center. The final anchor rating scale is shown in Table 3.

Secondly, two raters analyzed and scored the qualitative data obtained from the interviews of the 90 key informants independently based on the modified anchoring rating scale (see Table 3). As the anchored scales define each dimension by using a 9-point rating scale, with 1 representing the lowest levels of capability for that dimension and 9 representing the highest levels of capability, the raters finally scored each interview from 1 to 9 for the 6 dimensions, respectively. In order to ensure the reliability of the rating process, discussions of the items or phrases were made when disagreement occurred, and a third rater was invited to repeat the scoring until reaching a consensus.

Finally, we summed the consensual scores of the six key informants in every community and averaged them to calculate a final score for each dimension. The overall capability level of a community was determined by calculating the average scores of the six dimensions, and the overall level of each dimension for all communities under study was obtained by averaging the dimensional score of every individual community. All the averages were rounded down to generate whole numbers, which were used to identify the overall as well as specific dimensional levels of community capability.

## 4. Results

### 4.1. Sample Characteristics

The majority of the key informants were female (N = 52), accounting for about 58% of the total 90 interviewees. As for occupation, about 40% of the respondents (N = 37) were secretaries of the community neighborhood committees, who were regarded as the formal or official heads of the urban communities in China. The second largest number of the respondents were the volunteers in COVID-19 prevention (N = 24). There were also 18 community elites and 11 medical workers participating in the interviews. Moreover, the age of most of the respondents was 30 to 50 years, with the oldest being 54 years old. More than 50% of the key informants have lived in their communities for more than 5 years.

### 4.2. Overall Level of Community Capability

Figure 1 and Table 4 show the final community capability scores and levels of each dimension, respectively. On the whole, we found that the average capability score (ACS) of the total 15 communities was 4.97, which means that the overall level of community capabilities to prevent and control the COVID-19 pandemic in the city of Shenyang was located at the stage of preparation. Furthermore, the capability scores of the 15 communities ranged from 4.28 to 5.61, which indicates that their levels were mainly distributed in the stages of preparation, preplanning, and initiation.

Specifically, 26.67% of the evaluated communities were in the initiation stage, which demonstrates that a quarter of the communities in the city had taken some measures or considerable efforts to control and prevent the pandemic, and some active community members began to participate in the relevant programs. About 20% of the communities stayed in the preplanning stage, which means that there was recognition of the seriousness of the COVID-19 pandemic and the need to take some actions, but there were no concrete plans about how to control and prevent it. Moreover, no efforts or specific plans had been made to deal with the epidemic. More than 50% of the communities were found in the preparation stage, indicating that more than half of the communities were planning to solve the problem, which means that active leaders began to take some actions or formulate some schemes to fight against the disease and various resources were ready to put into use.

### 4.3. Overall Average Level of Each Dimension

Figure 2 displays visually the average scores for each dimension of the community readiness. Leadership, with an average score of 5.86, received the highest score among all the six dimensions. This puts leadership in the stage of initiation, demonstrating that the community leaders were actively running and supporting plans for combating the COVID-19 epidemic. This result was easily understood in the Chinese context because the whole country from the head of the state to the head of the communities all attached great importance to the prevention and control of the COVID-19 pandemic. Community attachment had the second highest mean score (5.20) of all the dimensions, placing it at the stage of preparation. This means that some community members felt that they had the responsibility to participate in the fight against the COVID-19 pandemic and tried best to make their contributions. Community knowledge of efforts and community knowledge about the issue received average scores of 5.09 and 4.87, respectively, both of which are in the stage of preparation. The evaluation results indicate that general information on COVID-19 was available in the communities and some community members had basic knowledge of the pandemic. In addition, the community members also had general knowledge of the plans, policies, emergency management projects of COVID-19 prevention and control, and the leaders and the people involved in combatting the pandemic. The overall average scores of community resources (4.36) and community efforts (4.43) were the lowest among all dimensions, indicating the preplanning stage. The results demonstrate that most of the evaluated communities were facing the dilemma of insufficient resources and inadequate efforts in the prevention and control of COVID-19.

### 4.4. Specific Levels of Each Dimension for the Communities

First, regarding the dimension of community knowledge about the issue, seven communities were at the stage of preparation, three communities were at the initiation stage, while one community was at the institutionalization stage. The remaining four communities were at the preplanning stage. The assessment results reveal that there was a significant difference between the communities in the level of knowledge about COVID-19. According to the interviews, members in some communities were well informed about COVID-19 and had some information about its causes and consequences through communities’ bulletin boards, WeChat groups, and government notices. However, residents in other communities were ill-informed. As a leader from community F said, “the publicity of COVID-19 was inadequate, especially in communities with large aging population, they don’t use the smartphones or internet”.

Second, in terms of community efforts, about half of the communities were at the stage of preparation while the other half were at the preplanning stage, which showed that there were no significant differences in the communities’ efforts, including preparing health emergency schemes, making emergency exercises, popularizing knowledge of tackling public health emergencies, and improving infrastructure construction, to prevent and control the COVID-19 pandemic. Nonetheless, some community efforts were irregular and temporary. As an informant of community G addressed in the interview, “In fact, emergency exercise is not often made and equipment is inadequate, especially for pandemic prevention and control”. In addition, most of the key informants interviewed noted that the lack of grant funding and experts were barriers for communities to make efforts.

Third, concerning the community knowledge of efforts, the levels of this dimension for all communities assessed were higher than that of the community efforts. Eleven communities were at the level of preparation and four communities were at the initiation stage, indicating that there were slight disparities in the communities’ level of knowledge of efforts. The evaluative result also means that community members in most of the communities had general knowledge of the efforts but lacked specific knowledge as well as in-depth understanding of the local efforts. Some key informants described the difficulties to acquire official and up-to-date information regarding the specific plans, policies, and emergency management programs of the COVID-19 prevention and control. As an informant from community L suggested, “further information of the efforts should be widely disseminated in the communities”.

Fourth, regarding the dimension of leadership, four communities were at the level of institutionalization, seven communities were at the stage of initiation, and the remaining four were at the preparation level. The evaluative results revealed that almost all the communities had a higher level of leadership than their level on the five other dimensions. The reason for this result might be that the leadership was very identifiable in the Chinese community context. They were composed of leaders in the community neighborhood committees, community self-government organizations, professional service organizations, and intermediary agencies. They made great efforts on preventing and controlling COVID-19. Most of the key informants interviewed stated that community leaders provided direct and enormous support in the prevention and control of COVID-19, such as distributing resources quickly, organizing staff orderly, and serving the residents wholeheartedly.

Fifth, with respect to the community resources, six communities stayed at the stage of preparation, while nine communities remained in the preplanning stage. The empirical results show that most of the communities had the lowest level of resources among all the dimensions. A possible explanation might be that China had the largest population and the Chinese government attached greatest importance to the prevention and control of the pandemic, which required a great many of financial, human, material, and many other resources to control the spread of COVID-19. In contrast, the various resources needed for preventing the COVID-19 emergency were limited. According to the interviews, the government appropriation and donations from the Red Cross Society were not sufficient to meet financial requirements. The number of community volunteers was also insufficient, with less than 30 volunteers in some communities.

Finally, regarding the dimension of community attachment, four communities were at a higher level of initiation, eight communities were at a medium level of preparation, and three communities were at a lower level of preplanning. The analytical results disclose a big difference in the level of various communities’ attachment. Furthermore, the level of community attachment for most of the communities was higher than the overall level of the community capability, which illustrates the higher level of community attachment than other dimensions. This result corresponded to the fact of Chinese community members’ high sense of participation and high level of involvement in the pandemic prevention and control.

## 5. Discussion

This study applied a modified framework of the CRM for the first time to evaluate community capability to prevent and control the COVID-19 epidemic in the Chinese context. The assessment results demonstrate that the overall capability levels of all the communities evaluated in this research were at the stages of preplanning, preparation, or initiation, which can be together called the “intermediate stages” group in the spectrum of the readiness model, according to Kelly et al. [63]. Apart from identifying the overall level of each community’s capability, we also assessed and presented the levels of the six different dimensions, on which community capability was based. The research findings of the present study not only have important theoretical contributions but also show some practical implications to inform policymakers in promoting community capabilities in the prevention and control of COVID-19 and other epidemics.

Firstly, the CRM has been widely used to assess community readiness levels in different countries, such as Australia, the USA, the UK, and other European countries [37,40,47,64]. This study contributed to prior studies by employing the model in the Chinese community context. Moreover, previous research adopting the CRM mainly focused on evaluating community readiness of general prevention programs, such as drug and alcohol [65], obesity [66], violence [67], and healthy lifestyles [68]. To the best of our knowledge, only Adane et al. used the model to assess the level of community readiness for COVID-19 pandemic prevention, which they assessed in the Awi Zone of northwest Ethiopia [32]. Thus, our study also helps to extend the application of the CRM to the evaluation of community capability in the prevention of a public health emergency.

Secondly, the CRM was modified in this study by substituting community attachment for the community climate dimension, in order to improve the appropriateness and applicability of the model in the context of Chinese community. Community attachment mainly refers to community members’ sense of belonging to their community. In other words, it emphasizes the emotional ties that community residents have to their communities. The final results illustrate that community attachment demonstrated a higher level than other dimensions of community capability except leadership. According to the interviews with key informants, the emotional and psychological ties between the residents and their community could activate residents’ willingness to devote various resources and take positive actions to support relevant programs in the prevention and control of the COVID-19 epidemic. This result was consistent with the findings of Peterson and Reid [69]. In addition, we refined a community readiness evaluation index to make it suitable for the assessment of Chinese community capability based on the anchor rating scale developed by the Tri-Ethnic Center, which provides guidance for the operationalization of the CRM in the Chinese context and promotes the application of the model in a wider range of subject areas and issues in China [70].

Thirdly, the number of evaluated communities in most previous studies was less than ten, and one third of prior studies focused on only one community [36]. This research made further contributions by assessing both the overall capability level of the fifteen communities as a whole and the individual level of capabilities for each community. Moreover, the average level of each dimension in the modified model was also reported for the fifteen communities, demonstrating the overall stages of the six dimensions. Thus, this study provides a comprehensive evaluation of community capabilities, by not only regarding the fifteen communities as a geographically larger community, namely the urban city, but also focusing on the different levels of capabilities to prevent and control COVID-19 between each individual community.

Fourthly, the analytical results of this study indicate that leadership is located at the highest level among all the six dimensions to measure community capability of preventing and controlling the COVID-19 pandemic in China. This result corresponds with the findings of Coroiu et al. [71] and Kostadinov et al. [72]. However, some previous studies found that the dimension of community efforts received the highest scores in both Western settings, such as the UK and the USA, and in the Middle East context, such as Iran [73]. A possible explanation for the different results may be that community leaders in China played an irreplaceable and vitally important role in the process of combating COVID-19, through following the leadership of the Communist Party of China, implementing epidemic prevention and control policy promulgated by the government, and mobilizing community residents to participate. They had authority, resources, and personal influence, which were all critical to improve the community capability level.

Fifthly, community resources and community efforts received the lowest scores in this evaluation, which demonstrates that these two dimensions could be regarded as the biggest weaknesses of the communities in the process of combating COVID-19 in China. Just as some scholars said, “resources are vital to any health intervention program’s success and they serve as potential indicator of future sustainability of the effort” [74]. According to the interviews with the key informants, relevant resources, including people, money, equipment, time, and space, were all insufficient to effectively deal with the pandemic. Specifically, limited financial support was a significant barrier when implementing some prevention programs. Infrastructures such as isolation space and hospital beds were in critical shortage, especially in the old and remote communities. There was also a huge shortage of medical workers and professional nursing staff. The finding that community resources were at the lowest level of capability conformed with existing research that inadequate budgets and expertise, and the underfunding of public health, were major and well-known problems [75,76,77]. Community effort was found to be the second lowest dimension, which contradicted prior studies on ordinary community prevention programs. The different results might be explained by the characteristics of health emergencies, including suddenness, complexity, and harmfulness, which usually made it difficult for communities to devote enough effort.

Finally, the evaluative results of this study also provided some important guiding suggestions and precise intervention strategies to promote the community capability of coping with COVID-19 and various other health emergencies to a higher level. As the results reveal, the levels of the community capability to prevent and control COVID-19 in Shenyang of China were in the stages of preplanning, preparation, and initiation. Therefore, we put forward a few policy suggestions based on the Tri-Ethnic Center Community Readiness Handbook and the evaluative results of the six dimensions for each community. Specifically, for the communities at the stage of preplanning, community members’ awareness of the various impacts of COVID-19 should be continuously enhanced through different forms of media; the engagement of various formal and informal community leaders in the preventive efforts should be increased via mobilization; and the emotional connection between residents and their communities should be reinforced through the provision of diversified forms of daily care and services and volunteer platforms.

For the communities at the stage of preparation, more local data about COVID-19 should be collected and made available for community residents; community key leaders and influential people should be motivated to mobilize community residents to provide more supports for the prevention and control of the pandemic; the effectiveness of preventive policies and programs should be evaluated; and community surveys and public forums should be conducted to solicit new preventive strategies from community leaders and community members. For the communities at the stage of initiation, professional in-service training of the prevention and control of the pandemic should be conducted; publicity efforts regarding the COVID-19 prevention should be further promoted; the progress of the epidemic prevention and control should be continuously updated; evaluation of the various preventive efforts should be increased; interviews with community members should be conducted to obtain their comments about improving the prevention strategies; and more resources including money and people should be invested.

## 6. Conclusions

This study evaluated the community capability to prevent and control the COVID-19 pandemic in Shenyang, the capital city of Liaoning province in Northeast China. The evaluation adopted a modified framework of a CRM by removing the dimension of community climate and adding a new dimension of community attachment. The assessment results demonstrate that the overall level of community capability in the pandemic prevention and control in the city of Shenyang was at the stage of preparation. The specific levels of the selected fifteen communities ranged from the stages of preplanning to preparation to initiation, showing a moderate level of difference. Regarding the level of each dimension, the levels of community knowledge about the issue, leadership, and community attachment exhibit significant disparities among different communities, while the levels of community efforts, community knowledge of efforts, and community resources show slight differences. In addition, leadership shows the highest overall level among all the six dimensions, followed by community attachment and community knowledge of efforts. Community resources present the lowest level, closely followed by community efforts.

Although this study has important theoretical contributions as well as practical implications, there are still some limitations that need to be emphasized. The most important limitation is that the evaluation of the community capability levels relied on the interviews with key informants, mainly including community neighborhood committee leaders, community workers, and community elites, who might overstate their roles and performance and underestimate the performance of others. Future studies could include a broader range of community informants. We also suggest that a combination of qualitative and quantitative approaches be used in future community capability assessment, in order to reduce the impact of subjectivity inherent in the key informant interviews. Another limitation is that the evaluative result of the overall level of community capability in Shenyang might be biased due to the limited number of selected communities and the lower administrative level of key informants in this research. It is suggested that more different types of communities be included in future analysis. Moreover, community key informants at a higher administrative level, including the sub-district, urban district, and the city, could be incorporated to obtain a better understanding of the overall level of community capability in the city. A third limitation of this study lies in its failure to analyze the impact of China’s COVID-19 policy on community capabilities to deal with public health crises. While policy formulation and implementation are crucial in strengthening community capacity, the influencing mechanism of the policies is worth further exploration. The last limitation is that the present study estimates the overall average level of community capability in Shenyang by randomly selecting 15 communities at the jurisdictional level of community neighborhood committee, rather than focusing on a single community. Although this research design has some theoretical contributions, it fails to analyze the specific characteristics of individual communities. An in-depth case study to evaluate a single community might provide more targeted intervention strategies for policymakers to enhance the level of community capability to manage public health emergencies.

## Figures and Tables

**Figure 1 ijerph-20-03996-f001:**
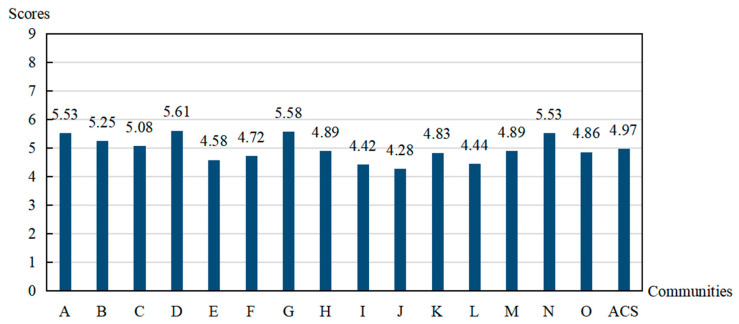
Community capability level.

**Figure 2 ijerph-20-03996-f002:**
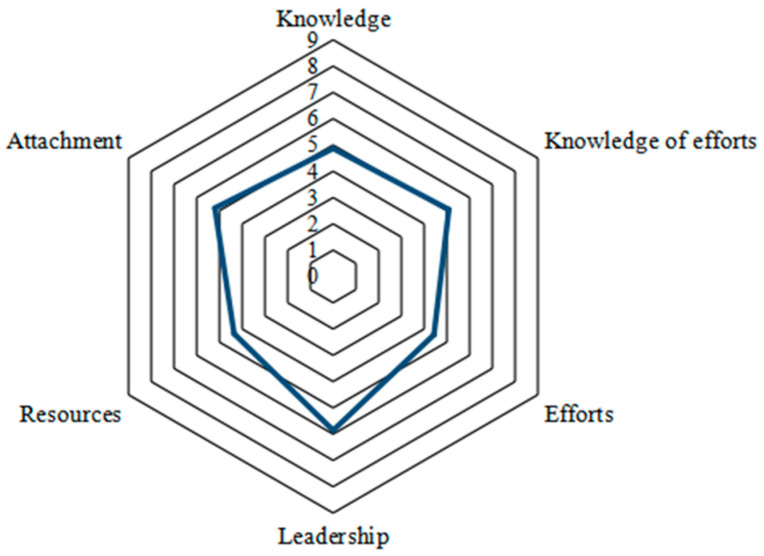
Community capability level of each dimension.

**Table 1 ijerph-20-03996-t001:** Stages of community readiness/capability.

Stage of readiness		Description	Scores	Capability Level
Stage1	Community tolerance	Communities and leaders have no awareness of the problem (or it may truly not be an issue).	1	Low
Stage2	Denial/Resistance	Few residents recognize that the issue is a concern, but there is little recognition that it might be occurring locally.	2	…… ……
Stage3	Vague awareness	The majority feel that there is a local concern, but there is no immediate motivation to do anything about it.	3	
Stage4	Preplanning	There is clear recognition that something must be done, but no efforts or specific plan has been done.	4	
Stage5	Preparation	Active leaders begin planning in earnest, community climate offers modest support of efforts, various community resources are ready to put into use.	5	
Stage6	Initiation	Activities are underway, and active community members begin to participate in the plan.	6	
Stage7	Institutionalization/ Stabilization	Activities are supported by administrators or community policymakers, resources, personnel and policies are fully equipped.	7	
Stage8	Confirmation/ expansion	With good results from the efforts, the community is prone to scale up its efforts and evaluate the project experience.	8	
Stage9	Professionalization	The community has sophisticated understanding of the problem, highly trained staff are in place. Effective evaluation can be done during this stage.	9	High

Adapted from Edwards et al. (2000) [35].

**Table 2 ijerph-20-03996-t002:** Interview outline.

A. Existing Community Efforts and B. Community Knowledge of the Efforts
Q1: What efforts are currently available in your community that relate to combating the COVID-19 epidemic? For example, rules, regulations, programs and so on. Please explain.
Q2: How long have these policies or rules been operating in your community?
Q3: Who are served by these efforts?
Q4: How are these efforts implemented?
Q5: Do you know what emergency infrastructure has been built or planned to be constructed in your community? Can you give an example?
Q6: If a person does not follow the relevant policies about COVID-19 prevention and control, will he or she be punished?
Q7: Is there a need to expand existing efforts about the COVID-19 prevention and control? If no, why not?
Q8: How are existing efforts viewed by the community members?
Q9: Are there plans to expand or develop new activities to fight against COVID-19?
Q10: Using a scale from 1 to 10, how aware are people in the community of these efforts, with one being no awareness and ten being very aware? Please explain.
**C. Leadership**
Q11: Do you know who the leaders in your community are?
Q12: Who or which group is mainly responsible for epidemic prevention and control?
Q13: Are there some informal leaders whose opinions are respected or some influential people who were contacted when the COVID-19 broke? Explain how they become “leaders”?
Q14: Do the leaders attach great importance to the prevention and control of the COVID-19 epidemic?
Q15: Are these leaders in your community involved in formulating and implementing prevention measures? Please explain how and what measures they are involved in?
Q16: Are there any community organizations involved in the COVID-19 prevention and control?
Q17: Using a scale from 1 to 10, how much of a concern is the issue of COVID-19 prevention and control to the leadership in your community, with one being not at all and ten being of great concern? Please explain.
**D. Community Knowledge about the Issue**
Q18: What public health emergencies have you experienced or learned about, and can you give some examples?
Q19: Do you know who is more vulnerable to the COVID-19 epidemic?
Q20: Do you know the symptoms of the COVID-19 epidemic?
Q21: Do you know the impact of the COVID-19 epidemic on our life and productivity?
Q22: Has your community set up some channels to understand and prevent the COVID-19?
Q23: Is there information available on the extent of the COVID-19 epidemic spreading? If yes, from whom?
Q24: Do the authorities release any information about the COVID-19 outbreak? Is it timely?
Q25: Using a scale from 1 to 10, how much do community members know about COVID-19 in your community, with one being not at all and ten being a lot? Please explain.
**E. Community Resources**
Q26: Do you know the source of community funds that are mainly invested in the prevention and control of COVID-19?
Q27: When COVID-19 broke out, how did the community obtain relief goods, funds, or professional relief talents? Is the community taking full advantage of them?
Q28: What resources has the community invested in combating COVID-19?
Q29: What is the community’s attitude about supporting prevention efforts with either people, money, time, or space?
Q30: What is the level of expertise and training among those working toward prevention of the COVID-19 epidemic?
Q31: Whom would an individual infected with COVID-19 turn to first for help, and why?
**F. Community Attachment**
Q32: What is the size of your community? How long have you worked (lived) in it? Are you satisfied with the current community efforts to prevent and control COVID-19?
Q33: Do most community members participate in one or more non-governmental organizations or clubs?
Q34: Are there any nongovernment organizational volunteers involved in the prevention and control of the COVID-19 epidemic?
Q35: When community members are in trouble, do they turn to their communities for help?
Q36: Have you ever made suggestions to community neighborhood committees to improve the community capability level of COVID-19 prevention and control?
Q37: Are community members cooperating well in the fight against the COVID-19 epidemic?
Q38: Do community members actively participate in the community volunteer service for epidemic prevention and control?
Q39: What are the factors that affect the participation or cooperative behaviors of the community members?
Q40: Using a scale from 1 to 10, please score how close the connection is between community members and the community, with one being not at all and ten being very close.

**Table 3 ijerph-20-03996-t003:** Anchor rating scale.

A. Existing Community Efforts
1. No awareness of the need for efforts to combat the COVID-19 epidemic in any capacity.
2. No efforts to prevent and control the COVID-19 epidemic. For example, plans, policies, etc.
3. A few individuals recognize the need to initiate some types of efforts, but there is no immediate motivation to do anything.
4. Some community members have met and begun a discussion of developing community efforts to combat the COVID-19 epidemic.
5. Policies and contingency items responding to the COVID-19 emergency have been put on the agenda, and community workers are also being trained.
6. Policies and resources are being prepared and the COVID-19 emergency response programs are being implemented.
7. Some plans, policies, emergency management programs have been in operation for several months and will continue to operate, but no new programs are expected.
8. Several different plans, policies, emergency management programs are running in the community with a wide range. New efforts are being developed based on evaluation data.
9. The community evaluates the effectiveness of different plans, policies and emergency management projects, and makes continuous improvements based on the evaluation results.
**B. Community Knowledge of the Efforts**
1. Community has no knowledge of the need for efforts to prevent and control the COVID-19 epidemic.
2. Community has no knowledge about efforts to prevent and control the COVID-19 epidemic.
3. A few community members heard about the COVID-19 emergency contingency plans, policies, emergency management programs, but have no real information on what they do and how.
4. Some leaders actively seek information about the plans, policies and emergency management programs of the COVID-19 prevention and control.
5. The community members have general knowledge of the COVID-19 emergency management plans, policies and projects, the leaders and the people involved.
6. An increasing number of community members have knowledge of local efforts and are trying to increase the knowledge of the general community about these efforts.
7. There is evidence that the community has specific knowledge of local efforts in the process of COVID-19 prevention and control, including contacting persons, training of staff, clients involved, etc.
8. Most community members have an in-depth understanding of the community emergency management plans, policies and projects, and have professional knowledge about the COVID-19 emergency.
9. Community has knowledge about program evaluation data on how well the different local efforts are working and their benefits and limitations.
**C. Leadership**
1. Leadership has no awareness about the problem of COVID-19 epidemic.
2. Leadership believes that COVID-19 epidemic is not a problem in our community.
3. Leader(s) recognize(s) the need to do something to prevent and control the COVID-19 epidemic.
4. There are identifiable leaders who start trying to do something, such as a meeting to discuss the COVID-19 epidemic prevention and control.
5. Leaders are part of a committee or group that addresses the prevention and control of the COVID-19 epidemic.
6. Plans for preventing and controlling the COVID-19 epidemic are running and supported by the committee leaders, but there is lack of cooperation.
7. Community leaders are strong supporters of plans for preventing the COVID-19 epidemic and are considering resources available for self-sufficiency.
8. Community leaders support a variety of emergency response programs, with staff well-trained, community leaders and volunteers actively involved. Independent assessment teams are running.
9. Leaders are continually reviewing evaluation results of the efforts and are modifying support accordingly.
**D. Community Knowledge about the Issue**
1. Few community members consider the COVID-19 epidemic to be a problem or that it would cause problems.
2. Only a few community members have some knowledge about COVID-19, while many community members have misconceptions about the epidemic.
3. A few community members have basic knowledge of the COVID-19 epidemic and recognize that some people here may be affected by the epidemic.
4. Some community members have basic knowledge (causes, consequences, signs and symptoms) and recognize that the COVID-19 epidemic occurred locally, but access to information is lacking.
5. Some community members have basic knowledge of COVID-19. General information on COVID-19 is available.
6. A majority of community members have basic knowledge of COVID-19, including modes of transmission, means of prevention, understanding of high-risk groups and behaviors. Specific local data on COVID-19 is available.
7. Community members have knowledge of, and access to, detailed information about local prevalence of COVID-19.
8. Community members have substantial knowledge about the prevalence, causes, risk factors and consequences of the COVID-19 epidemic.
9. There is detailed information on local and national changes about the COVID-19 epidemic; and community members know a lot about the effectiveness of local prevention measures.
**E. Community Resources**
1. There are no resources available for the prevention and control of the COVID-19 epidemic.
2. There are very limited resources available that could be used for the COVID-19 prevention and control. There is no action to allocate these resources to the preventive efforts.
3. There are some resources that could be used for the COVID-19 prevention and control. There is little action to allocate these resources to the preventive efforts.
4. There are some resources (individuals, organizations, and/or space) identified that could be used for the COVID-19 prevention and control. Some community members or leaders have looked into or are looking into using these resources to prevent and control COVID-19.
5. The various resources needed to prevent and control the COVID-19 epidemic are known. Some community members or leaders are actively working to secure these resources, and funding proposals have been prepared, submitted and may be approved.
6. New resources have been obtained and/or allocated to support further efforts to fight COVID-19.
7. Much of the support comes from local sources, but they are uncertain and unsustainable. Community members and leaders are beginning to look at continuing efforts by accessing additional resources.
8. Diversified resources and funds are secured and efforts are expected to be ongoing. There is additional support for further preventive efforts.
9. There is continuous and secure support for programs and activities related to the COVID-19 prevention and control, and there are substantial resources for trying new preventive efforts.
**F. Community Attachment**
1. Community members feel no attachment with their community.
2. Very few community members feel attached to their community. Most community members think that preventing and controlling the COVID-19 epidemic is of non-relevance with them.
3. A few community members feel attached to their community and have a sense of responsibility to participate in the COVID-19 prevention and control but think that “there is nothing we can do”.
4. A few community members feel attached to their community and think that they should do something together to combat COVID-19, but don’t know what to do.
5. Some community members get close connections with their community, and they are planning to be united to combat the COVID-19 epidemic.
6. Some community members have close connections with their community and make some cooperative efforts to combat COVID-19.
7. Some community members feel highly attached to their community. They make cooperative efforts through a variety of channels to fight against the COVID-19 pandemic.
8. The majority of community members feel attached to their community. They trust the community leaders and devote cooperative efforts to support the COVID-19 pandemic prevention and control.
9. The majority of community members feel highly attached to their community. They have high sense of responsibility to participate and actively involved themselves in the cooperative efforts to fight against the COVID-19 epidemic. Community members and leaders trust and understand each other.

**Table 4 ijerph-20-03996-t004:** Community readiness scores.

Community	Knowledge of Issues	Efforts	Knowledge of Efforts	Leadership	Resources	Attachment	Average	Stage
A	5.67	4.67	5.83	6.67	4.83	5.50	5.53	Initiation
B	5.17	4.67	5.50	5.67	4.67	5.83	5.25	Preparation
C	4.50	4.50	5.17	5.50	5.00	5.83	5.08	Preparation
D	4.83	5.33	5.67	6.83	5.00	6.00	5.61	Initiation
E	4.50	3.83	4.83	5.83	4.00	4.50	4.58	Preparation
F	4.33	4.33	5.17	5.00	4.17	5.33	4.72	Preparation
G	6.00	4.83	5.50	6.83	4.67	5.67	5.58	Initiation
H	4.17	4.50	5.00	6.17	4.33	5.17	4.89	Preparation
I	3.67	4.33	4.83	5.33	3.83	4.50	4.42	Preplanning
J	3.67	3.83	4.50	5.17	3.67	4.83	4.28	Preplanning
K	4.67	4.50	5.00	5.67	4.17	5.00	4.83	Preparation
L	4.50	4.17	4.50	4.83	4.33	4.33	4.44	Preplanning
M	6.00	4.33	4.83	5.83	3.83	4.50	4.89	Preparation
N	6.67	4.33	5.17	6.67	4.83	5.50	5.53	Initiation
O	4.67	4.33	4.83	5.83	4.00	5.50	4.86	Preparation
Average	4.87	4.43	5.09	5.86	4.36	5.20	4.97	Preparation

## Data Availability

Data may be obtained from the corresponding author upon the consent of all the interviewees.

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
