# Peer review of "Evaluating Community Capability to Prevent and Control COVID-19 Pandemic in Shenyang, China: An Empirical Study Based on a Modified Framework of Community Readiness Model"

_ijerph, 2023, doi:10.3390/ijerph20053996_

Round 1
Reviewer 1 Report
A few minor English expression issues to note.
A relevant study that is of interest in the context of community resilience. Use of 'the' before COVID in many places needs to be revised. See attached for suggestions.
My only real issue is that the 9 stages outlined on p.3 need to be made clearer and restated with the results. Perhaps a table of these as they are highly relevant to the paper.

Author Response
Dear Reviewer 1,
Thank you for taking time to review our paper and giving us an opportunity to revise and resubmit. We sincerely appreciate your comments and suggestions. They are very helpful for us to revise the paper.
Please see the attached file for our detailed response to the review comments.

Reviewer 2 Report
This article seeks to evaluate the extent to which community-level bodies can address public health crises, especially in terms of COVID-19, in China. It modifies existing assessment frameworks to include a dimension of community capability that is pertinent to the Chinese context: community attachment. By seeking to understand community capability through a systematic, quantitative framework, this study provides a useful way to compare across communities as well as build evidence for the kind of governmental support needed by communities. However, some aspects of the study need to be clarified.
-- The study provides an interview questionnaire (Table 1) and an anchor rating scale (Table 2). It is not clear how these two are connected to one another. Further, which of these is being used to measure community capability and community readiness as depicted in Figure 1 and Table 3? The analytical steps connecting the interview questionnaire/anchor rating scale and the results need to be more clearly specified.
-- The research questions mentioned on lines 62-65 need to be more direct. For example, the sentence could be revised as: 'The main questions that this research addresses are (i) How can community capability be evaluated in China? and (ii) What are the levels of community capability for epidemic prevention and control in communities within Shenyang, the capital of Liaoning province?' There also needs to be some discussion of why Shenyang is a useful case study.
-- Iran is mentioned as having a 'western background' [lines 403-405] - this does not seem correct.
-- The term 'geographical community level [p.481] should be replaced by 'scale of administration'?
-- It would be useful if the article could end with some reflection of how change in China's COVID-19 policy might affect or not affect community capabilities to deal with public health crises. Were communities merely responding to government policy in terms of addressing COVID-19, or were they working to protect their members in the larger interests of the community?
Author Response
Dear Reviewer 2,
Thank you for your thoughtful review and constructive comments on our study. We sincerely appreciate your comments and suggestions. They are very helpful for us to revise the paper.
Please see the attached file for our detailed response to the review comments.

Reviewer 3 Report
1. The structure of the manuscript should be improved. The leading theory appears in the methods section, while it should appear in the literature review along with a more critical review and comparison between the several models on community capacity that are mentioned there. What are the advantages of each model and why was the community readiness model chosen? The discussion mainly repeats literature from the introduction instead of getting into a deeper understanding of the results, and there are also missing parts in the methods section (see following points).
2. The authors should better explain why community attachment and community climate are the same in this culture.
3. The first two pages of the introduction should be rephrased in a more modest way. We now know that the strict measures taken was less effective than described in the paper. Another example of the need to rephrase is "Facts have proven that these measures taken by communities effectively reduced the risk of the COVID-19 outbreaks". Which facts? What is the supporting evidence?
4. Are readiness and capacity/capability of the community synonyms? If so, this should clearly be stated.
5. Methods- Authors should indicate how ethical approval was received. This is always critical, especially with this research data collection of nonanonymous in-depth interviews.
6. Methods- it is hard to understand how the scoring was provided. What was the anchor scoring like? Some examples are missing and measures of quality assurance- interrater reliability.
7. Results- were there any significant differences between the dimension of readiness? Between communities? Were there any associations between the dimensions among themselves and with a general measure of readiness? Otherwise, it is less accurate to conclude (in the discussion) which of the components was more crucial than the others.
8. Discussion- the explanation of the lack of resources is nice and should be expanded. What were the missing resources? In which type of communities there were scarce resources and how they were related to the results? Etc.
Author Response
Dear Reviewer 3,
Many thanks for the constructive reviews on our manuscript. We sincerely appreciate your comments and suggestions. They are very helpful for us to revise the paper.
Please see the attached file for our detailed response to the review comments.

Reviewer 4 Report
The reviewed study presents a modified model for diagnosing the level of readiness of the commune community to prevent the COVID-19 pandemic, control its course and combat its effects. The described model is designed to diagnose local communities in the People's Republic of China. The assumed goals of the reviewed study are precisely reflected in the adopted title of the article. The authors verified the social usefulness of the model through empirical research. They showed its limitations and directions for improvement in order to use its advantages in diagnosing the readiness of societies living in cities.
In the opinion of the reviewer, the content of the article is logically arranged and relatively clearly shows the research method and its results. On the other hand, from the point of view of the social usefulness of the results of the study, conducting analyzes only in relation to average values can be considered a weakness of the article. They show general trends, but do not represent the areas of functioning of individual communities that need to be strengthened in order to improve the level of its resilience. According to the reviewer, the article should be supplemented with these issues. Another disadvantage of the analysis can be considered the lack of a quantitative scale adequate to the adopted nine levels of analysis. Without such a scale, assigning a mean value to a given level raises debate. In the opinion of the reviewer, the faulty assumption of the model is the use of nine levels of community readiness analysis. Such fragmentation for management practice is of little use, inter alia, due to the vagueness of the criteria and the degree of intensity of variants of the examined features. The use of only four/five levels of analysis can increase the degree of social usefulness of the proposed model. In order to improve the clarity of the presentation of the results, it is advisable to describe the coordinate axes in Figure 1.
The article is suitable for publication after applying the indicated corrections.
Author Response
Dear Reviewer 4,
Thank you very much for your time involved in reviewing the manuscript and your very constructive comments on our manuscript.
Please see the attached file for our detailed response to the review comments.

Round 2
Reviewer 3 Report
None